# Mechanism of Action and Efficiency of Ag_3_PO_4_-Based Photocatalysts for the Control of Hazardous Gram-Positive Pathogens

**DOI:** 10.3390/ijms241713553

**Published:** 2023-08-31

**Authors:** Emil Paluch, Alicja Seniuk, Gustav Plesh, Jarosław Widelski, Damian Szymański, Rafał J. Wiglusz, Martin Motola, Ewa Dworniczek

**Affiliations:** 1Department of Microbiology, Faculty of Medicine, Wroclaw Medical University, Tytusa Chałubińskiego 4, 50-376 Wroclaw, Polandewa.dworniczek@umw.edu.pl (E.D.); 2Department of Inorganic Chemistry, Faculty of Natural Sciences, Comenius University Bratislava, Ilkovicova 6, 842 15 Bratislava, Slovakia; plesch@fns.uniba.sk (G.P.); martin.motola@uniba.sk (M.M.); 3Department of Pharmacognosy with Medicinal Plants Garden, Lublin Medical University, 20-093 Lublin, Poland; jwidelski@pharmacognosy.org; 4Institute of Low Temperature and Structure Research, Polish Academy of Sciences, Okolna 2, 50-422 Wroclaw, Polandr.wiglusz@intibs.pl (R.J.W.)

**Keywords:** silver phosphate composites, hydroxyapatite, titanium dioxide, photocatalytic inactivation of bacteria, *Enterococcus*, *Staphylococcus*, mechanism of action, ROS, biofilm prevention

## Abstract

Silver phosphate and its composites have been attracting extensive interest as photocatalysts potentially effective against pathogenic microorganisms. The purpose of the present study was to investigate the mechanism of bactericidal action on cells of opportunistic pathogens. The Ag_3_PO_4_/P25 (AGP/P25) and Ag_3_PO_4_/HA (HA/AGP) powders were prepared via a co-precipitation method. Thereafter, their antimicrobial properties against *Enterococcus faecalis*, *Staphylococcus epidermidis*, and *Staphylococcus aureus* (clinical and reference strains) were analyzed in the dark and after exposure to visible light (VIS). The mechanism leading to cell death was investigated by the leakage of metabolites and potassium ions, oxidative stress, and ROS production. Morphological changes of the bacterial cells were visualized by transmission electron microscopy (TEM) and scanning transmission electron microscopy with energy-dispersive X-ray spectroscopy (SEM EDS) analysis. It has been shown that Ag_3_PO_4_-based composites are highly effective agents that can eradicate 100% of bacterial populations during the 60 min photocatalytic inactivation. Their action is mainly due to the production of hydroxyl radicals and photogenerated holes which lead to oxidative stress in cells. The strong affinity to the bacterial cell wall, as well as the well-known biocidal properties of silver itself, increase undoubtedly the antimicrobial potential of the Ag_3_PO_4_-based composites.

## 1. Introduction

Due to the growing resistance of microbes to antibiotics and disinfecting agents, as well as the resulting public health concerns, it has become necessary to develop new, effective antimicrobial agents. Those agents that would be durable, stable, and possessed of consistently high activity during the entire period of use are of special interest [1,2]. In hospitals, UV light is usually used for periodic sterilization of rooms and surfaces. However, despite its effectiveness, it only works for a short time. They can certainly be a supplement or an alternative to other disinfectants (alcohols, phenols, aldehydes, or quaternary amine salts) [3].

Gram-positive bacteria such as *Staphylococcus* or *Enterococcus* are a significant problem in the hospital environment not only because of their multidrug resistance, but also because they are able to contaminate usable surfaces for a long time. In favorable conditions of the microenvironment, those bacteria are prone to produce a biofilm in a short time, which causes a constant threat to hospitalized patients, increasing the risk of infections [4]. Therefore, an important element is to prevent this process by searching for compounds that will be able to permanently reduce the number of pathogens in the hospital environment, limiting the possibility of severe infections, while at the same time being safe for the patient [5,6].

Photocatalytic oxidation using visible light may be a novel approach to solving this urgent and important problem. Nanosized inorganic and organic particles used in photocatalytic processes seem to be a highly effective weapon against microorganisms, especially multi-drug resistant strains. Nanoparticles easily penetrate the interior of the cell with micro dimensions, through transport channels located in its sheaths. Depending on their nature, they disrupt the functioning of individual cell structures, thus leading to the death of the microorganism [7]. Due to their favorable chemical parameters and the rarely observed induction of resistance, inorganic materials containing metals such as silver and titanium are of special interest to researchers [8].

One of the commonly used traditional photocatalytic materials is titanium dioxide (TiO_2_) due to its great features like low cost, non-toxic as well reusability [9]. However, TiO_2_ can be active as a photocatalyst in the range of UV light (only 5% of the solar radiation spectrum), because it exhibits a relatively large bandgap of 3.2 eV [10,11]. In turn, the valence band level of silver orthophosphate is lower than titanium dioxide and holes (h^+^) photogenerated in the valence band of Ag_3_PO_4_ can be transferred to the TiO_2_ valence band and are responsible for the initiation of different oxidation reactions [12].

Ag_3_PO_4_ is a semiconductor photocatalyst with the ability to effectively suppress Gram-positive and Gram-negative bacteria, as well as fungal hyphae [11]. The silver orthophosphate has great antibacterial properties, but on the other hand, Ag_3_PO_4_ micro- and nanoparticles are affected by poor solubility, and the photoreduction of silver ions (Ag^+^) to metallic silver (Ag^0^) is prone to occur in the case of absence of electron acceptors, which is responsible for the reduction of its structural and photocatalytic stabilities [11]. The ideal combination of Ag_3_PO_4_ with TiO_2_ in the form of heterostructure enables higher photocatalytic and antimicrobial activities under the same conditions.

The mechanism of action of this type of compound against microorganisms is usually multifactorial and detailed studies are often necessary for the correct interpretation of obtained results. One of the very important signaling processes in the cell is the normal level of reactive oxygen species (ROS). However, certain factors (e.g., photocatalysis, nanoparticles or other compounds) can disrupt homeostasis and lead to a high level of oxidative stress which will usually cause irreversible intracellular damage and apoptosis [13,14]. Photocatalysis materials, where electrons from the conduction band recombine with holes from the valence band of photocatalytic materials, create ROS (including intercellular ROS) such as superoxide anions (O_2_ ^• −^), hydroxyl radicals (^•^ OH), and singlet oxygen (^1^O_2_) [15,16]. Reactive oxygen species can lead to the beta-oxidation of fatty acids found in cell membranes and increase their permeability and disruptions. In this way, they can lead to a disturbance of the cell’s ionic balance (the leakage of Na^+^ and K^+^ ions) [17]. Microcomposites can also act directly on the surface of bacteria and lead to alkylation of surface proteins. Also, increased levels of free radicals in the microenvironment can lead to mitochondrial damage and further increase oxidative stress and the risk of severe DNA damage (e.g., microcracks) with consequent apoptosis [18,19].

In our work, we would like to draw attention not only to the effect of photocatalytic microbial inactivation by composites of silver phosphate with TiO_2_ (P25) and hydroxyapatites (HA), but above all to its molecular intracellular mechanism (in particular on the generation of iROS, DNA release ions leakage) in the model of *E. faecalis* VRE 037 clinical strain, which is a nosocomial pathogen. The practical use of this knowledge may reduce the risk of nosocomial infections, especially those resulting from the presence of pathogens in the patient’s immediate environment.

## 2. Results and Discussion

The results in the first stage of research show screening studies on the inactivation of Gram-positive bacteria by Ag_3_PO_4_-based materials with the use of reference strains and multidrug-resistant clinical strains of the *Staphylococcus* and *Enterococcus,* bacteria of great importance in the hospital environment. In further studies, *E. faecalis* VRE 037 was selected as one of the most important and difficult to eradicate the clinical model organism which is useful for the study of the mechanism of action of the tested materials.

The results of presented studies aimed at assessing the susceptibility of the strains *E. faecalis* 37VRE, *E. faecalis* ATCC 29212, *S. aureus* ATCC BAA 1556 (USA 300), and *S. epidermidis* P36 to photocatalytic inactivation with the use of the studied microcomposites are shown in Figure 1. To evaluate the antimicrobial activity we used logarithmic reduction (log^10^ CFU/mL). As shown in Figure 1, exposure of bacteria only to VIS light for 30 and 60 min, without the use of photocatalysts, did not inhibit the survival of bacteria (control). Similarly, the samples were not exposed to VIS light, with the presence of AGP, AGP/P25, and HA/AGP used at concentrations of 750 and 1500 mg/mL and affecting bacteria for 30 min and 60 min did not cause a significant reduction in the growth of individual strains. However, significant differences in the survival of bacteria appeared after their exposure to photocatalysts combined with VIS irradiation. Photocatalysis conducted for 30 and 60 min with the participation of AGP, AGP/P25, and HA/AGP used at the concentration of 750 mg/mL and 1500 mg/mL effectively destroyed *E. faecalis* 37VRE and *S. aureus* ATCC BAA-1556 (USA 300) cells (decrease from 6 log CFU/mL to 0). One log reduction unit equates to a 90% decrease in the number of bacteria (Figure 1). However, in the case of strains *E. faecalis* ATCC 29212 and *S. epidermidis* P36, complete eradication with both concentrations of the tested materials occurred only after 60 min of VIS exposure (survival decrease to 0). Reducing the time of exposure to 30 min led to an increase in the survival of these strains (Figure 1). The research by Seo et al. and Bonetta et al., where strong activity against *S. aureus* (3 log decrease) was demonstrated for the surfaces coated with TiO_2_ (P25) [7,20], the antimicrobial activity of P25 material was also demonstrated on orthopedic implant surfaces where they limited the growth of *E. hirae* [21]. The results of our studies correspond with the antibacterial activity of Ag_3_PO_4_/TiO_2_ heterojunctions against the reference strain of *S. aureus* (ATCC 25933). In the experiment conducted by Lyu et al. MBC value (with 1:2 ratio Ag_3_PO_4_ to TiO_2_) was 20 mg/mL [11]. In another experiment, Ag_3_PO_4_/TiO_2_ nanocomposite fibers showed excellent performance for the inactivation of *S. aureus* (ATCC 29231) in comparison to the pristine TiO_2_ nanofibers. Moreover, a clear inhibition ring around the Ag_3_PO_4_/TiO_2_ nanocomposite fibers compared to TiO_2_ nanofiber indicates their efficiency as an antimicrobial agent [22]. Ternary heterostructures composed of P25/Ag3PO4/graphene oxide enhanced solar photocatalytic degradation of reference strains of *S. aureus* (MIC /MBC values were 6.25/12.5 ppm) [23]. The excellent antibacterial activity against *S. aureus* is characteristic of a combination of hydroxyapatite with silver orthophosphate. For example, Ag_3_PO_4_-loaded hydroxyapatite nanowires completely inhibited the growth of *S. aureus* at a concentration of 15 mg/mL. Present data concern the Ag_3_PO_4_-HAP-25 composition (25 describes nm concentration of AgNO_3_ used for the preparation of composition) [24]. Similar results concerning the activity of silver phosphorate/hydroxyapatite composition against the reference strain of *S. aureus* were obtained by Elyacoubi et al. [25]. In both studies the pure hydroxyapatite (control) was inactive. Very interesting is the fact that, generally, the composition of Ag_3_PO_4_ with other semiconductors like TiO_2_ or hydroxyapatite is more active against Gram-negative bacteria (e.g., *E. coli*) than Gram-positive. This tendency could be explained by the fact that Gram-positive bacteria (*S. aureus*) have thick peptidoglycan layers [24]. This confirms that the tested composites can be used as antimicrobial surfaces in the pharmaceutical or medical equipment and environment. The great value of our work is the large scope of research both in terms of the diversity of compositions containing silver phosphate and their activity against four strains of Gram-positive bacteria. Most of the research concerns one composition with photocatalytic properties and its activity against one or two bacterial strains.

The next stage of this study was to determine a possible mechanism of action of photo-activated investigated materials on a selected clinical model strain of *E. faecalis* VRE. For this purpose, the plate counting method (PCM) was supplemented with the analysis of cell viability in the presence of the tested materials using confocal laser scanning microscopy (CLSM), because it was noticed that some bacterial cells can aggregate on the surface of the tested materials. This part of the experiment allowed for obtaining much more information.

The CLSM analysis showed a strong decrease in the viability of *E. faecalis* on tested materials. In the top row (without exposure to VIS—“dark”), bacterial cells showing high viability similar to the negative control can be observed only under the influence of the AGP/P25 compound, as a partial appearance of dead cells can be noticed. Upon exposure to VIS (Xenon), all tested compounds showed a strong effect on the reduction of bacterial viability. A particularly large number of dead cells (98%) was observed under the influence of HA/AGP and AGP/P25 compounds, where it was also observed that dead cells strongly aggregate around the microparticles of the tested compounds. As a positive control, 1.5% hydrogen peroxide was used, which generated severe oxidative stress, where a 99% decrease in cell viability was observed (Figure 2). A similar effect under the influence of P25 was observed for *E. coli*, where a decrease in viability of up to 96% was obtained; however, it is assumed that Gram-negative bacteria may be more sensitive than positive bacteria [26]. Convergent results of studies with a high decrease in cell viability were observed against the biofilm of *S. aureus* and *E. faecalis* under the influence of various spatial forms of silver phosphate, as well as TiO_2_, although weaker activity was shown for P25 against *E. faecalis* [20,27].

In the next stage, we focused on the molecular mechanism activity of the tested compounds, because some of them may directly or indirectly disrupt the continuity of bacterial surface structures. For this purpose, we have tried to answer how much damage is caused to cells and what they can cause. Under the influence of the tested photocatalytic materials under irradiated (XE) and control conditions (non-irradiated (D), no leakage of macromolecular cellular components such as DNA and proteins was observed. Only bacterial cells under the influence of AGP/P25 biomaterial after irradiation (XE) showed increased leakage of DNA and protein (*p* < 0.05) (Figure 3A,B). These results suggest that we do not have (apart from the AGP/P25 compound) to act with large cellular damage and its complete destruction of bacteria. Therefore, we tried to determine whether the increased cell death rate may result from a disturbance in the ionic metabolism of bacteria. Significant differences were observed during the analysis of ion leakage from the cells of the tested bacteria. It was shown in the experiment that most tested compounds cause an increase 2 times in the leakage of potassium ions (K^+^) compared to the negative control, while its release increases under the influence of photocatalysis (*p* < 0.05) (Figure 3C). Lots of research, including studies in which Angelis et al. suggest that nanoparticles may be released from TiO_2_ impurities, which disturb the ionic balance of cells and lead to structural damage [28,29]. Then, we wanted to go a step further. The aim was to determine whether metal ions that do not occur naturally in the cell, such as Ag^+^ or Ti^2+^, can penetrate directly inside the bacteria. The experiment determining the content of metal ions (Ag^+^) and (Ti^2+^) in *E. faecalis* cells showed that all tested biomaterials cause significant penetration of metal ions into the cells (*p* < 0.05). Moreover, for the silver ion (Ag^+^), this process is particularly favored by the phenomenon of photocatalysis, where its increased content in bacterial cells is observed. The content of titanium ions (Ti^2+^) in bacterial cells was confirmed for AGP/P25 and was independent of the photocatalysis process (Figure 3D). It should also be emphasized that the cells were repeatedly cleaned of the tested materials, and, next, their density was standardized for each sample. Also in other studies, it was observed that the composites may lead to an increased release of metal ions (Au^+^; Cu ^2+^; Zn^2+^; Ag^+^; Ti^4+^), and it is possible that the phenomenon of photocatalysis may additionally intensify it, which may directly decrease the viability of *S. aureus* and *E. faecalis* [30,31]. In some cases, silver ions can reach as much as 12 µg/mL or more, depending on the microstructure of the test compound or composite and the composition of the culture medium [20,32]. Metal ions can cause disturbances in the continuity of cell membranes, including contributing to an increased outflow of potassium (K^+^), which, combined with the possibility of the release of nanoparticles or microparticles by composites (additional contact activity), may also lead to an outflow of DNA and proteins from bacterial cells, which leads to their death [14]. An increase in the cells’ permeability may also be an indirect effect of ions and compounds penetrating a cell, causing oxidative stress and inducing oxidation of lipids and sterols by the release of intracellular ROS (iROS) [33,34].

The subsequent step of the research on the mechanism of action was to determine whether the tested compounds promote the increase in intracellular oxidative stress levels in Enterococci. We used microscopic methods and we measured the fluorescence intensity to obtain reliable results. Fluorescence microscopy showed that all tested compounds lead to an increased level of oxidative stress in *E. faecalis* cells under the influence of photocatalysis. However, it should be noted that, in some cases, bacterial cells closely adhere to the tested biomaterial microparticles, giving the illusion of fluorescence of the tested compounds. Moreover, under the influence of the compound (AGP/P25), cells that were not irradiated (D) showed fluorescence, and after photocatalytic irradiation (XE), mostly cellular debris and low fluorescence were observed in the field of view. Measurement of the fluorescence intensity (a) (DCF) showed a different level of overall oxidative stress depending on the compound used. An increased level of fluorescence units was demonstrated by the compound (HA/AGP XE) (after photocatalysis), of which the highest level of fluorescence was observed under the influence of the compound (AGP/P25) both without (highest value) (D) and after photocatalysis (XE) (*p* < 0.005) (Figure 4). Further analysis involving fluorescence microscopy showed that the tested compounds also increased the production of superoxide anion radical (b) (DHE), where fluorescence was observed under the influence of almost all compounds. Again, it was observed that only non-irradiated cells were fluorescent under the influence of the compound (AGP/P25). Analysis of the fluorescence intensity level (b) (DHE) of the amount of superoxide radical anion production showed an increased level of cell fluorescence under the influence of photocatalysis (*p* > 0.005). Increased level of fluorescence was demonstrated by the compound (AGP XE) and high level of fluorescence (HA/AGP XE) (highest level) and (AGP/P25) (*p* > 0.005) (Figure 4). In the analysis of the ability to produce highly reactive ROS by fluorescence microscopy (c) (HAF) under the influence of all test compounds without photocatalysis (D), no fluorescent cells were observed, and single fluorescent cells were observed after photocatalysis. The exception was the AGP/P25 compound, where the photocatalyzed bacterial cells did not show fluorescence. Spectrofluorometric analysis of the level of fluorescence (c) (HAF) showed a slight but increased production of highly reactive ROS under the influence of compounds (AGP) and (HA/AGP) (*p* > 0.005), regardless of the photocatalysis process. It is noteworthy that under the influence of the compound (AGP/P25 D) (*p* > 0.005), an unusually high increase in the level of fluorescence was observed, but not fully corresponding to the microscopic image, which may be associated with a longer preparation procedure and complete cell disintegration (which is also confirmed by the results of protein and DNA leakage) (Figure 3A,B and Figure 4). However, many studies on AGP and P25 mention a strong generation of ROS under the influence of photocatalysis, but poorly characterize iROS in bacteria [27,35]. High values for the superoxide radical (O2 ^• −^) suggest that it is responsible for most of the damage to bacterial cells during photocatalysis. In the case of AGP/P25, there is also the production of highly reactive ROS, including hydroxyl radical anion (HO •), which leads to almost complete destruction of cells (decrease in fluorescence signal) [19]. This may indicate that extracellular oxidative stress leads to iROS and cellular damage, ranging from protein alkylation to lipid oxidation, leading to increased membrane permeability and further destruction, possibly including DNA strand breaks and cell lysis [36,37,38].

The analysis of ROS studies using fluorescence microscopy with numerous visible structures of biomaterials and the presence of metal ions inside bacterial cells (ICP analysis) gave a hint to make more precise microscopic images using TEM and SEM with EDS techniques. TEM analysis confirmed the influence of the photocatalysis process on morphological changes in *E. faecalis* cells, as well as the presence of nano and microparticles in their surroundings. Cells exposed to photocatalysis show an increased heterogeneous density of their cytoplasm. Moreover, nanoparticles can interact with their surface. Based on the observation of TEM images, a series of compounds of SEM EDS analysis were selected. This analysis confirmed that the tested HA/AGP and AGP/P25 compounds, apart from the release of microparticles, also have the ability to release nanoparticles. Interesting observations were made by the team of Singh et al., who showed that bactericidal activity may also depend on the spatial structure of Ag_3_PO_4_ crystals. Higher antibacterial activity was demonstrated for cubic over tetrahedral forms against both Gram-positive and Gram-negative bacteria [27]. The visualization of the silver content and the reference to carbon atoms allowed us to confirm that the tested material releases silver particles that can interact with bacterial cells (Figure 5). The presence of micro and nanostructures from compounds/composites in the environment of cells can lead to a killing effect as a result of indirect (ROS production) or direct interactions (direct contact with the surface of particles) [39]. Enterococci can defend themselves against these adverse factors by thickening the cell wall. However, while it is often effective in the case of large-molecule compounds, in the case of generating iROS, it may be and is insufficient [40,41]. The molecular mechanism of action of the tested compounds against Gram-positive bacteria based on photoinactivation of *E. faecalis* is shown in the diagram (Figure 6).

## 3. Materials and Methods

In our research, we used materials and methods that will be presented in this chapter.

### 3.1. Nanopowders

List of nanopowders used in our research:

Ag_3_PO_4_/P25 (AGP/P25); P25–TiO_2_ powder 75 wt.% anatase:25 wt.% rutile phases)Ag_3_PO_4_/HA (HA/AGP); HA–hydroxyapatiteAg_3_PO_4_ (AGP)HA

### 3.2. Preparation and Characterization of Photocatalysts

All materials (listed in Section 3.1) were prepared via a co-precipitation method according to our previous reports [42,43]. Briefly, 0.75 M aqueous solution of Na_3_PO_4_ 12H_2_O was added dropwise under vigorous magnetic stirring into 0.25 M aqueous solution of AgNO_3_, followed by 12 h stirring in the dark (to prevent the photo corrosion of the material) to obtain Ag_3_PO_4_. For HA/AGP and AGP/P25 (both P25 and HA were of analytical grade without further purification provided by Cambioceramics), P25 or HA was dispersed in deionized water in an ultrasonic bath. After, the solution of AgNO_3_ (0.25 M) was added to P25 or HA, respectively, and the suspension was stirred for 2 h. Finally, the Na_3_PO_4_ 12H_2_O solution was added into the suspension and stirred for an additional 2 h on a magnetic stir and 1 h in an ultrasonic bath, respectively. Subsequently, the material was washed several times using deionized water and dried at 80 °C for 24 h.

Materials were characterized in our previous reports [42,43] using X-ray Diffraction (XRD, PANalytical EPert PRO MRD diffractometer using Cu-Kα irradiation in Bragg–Brentano mode), Scanning Electron Microscope (SEM) with back-scattering electron (BSE) method, and Energy-dispersive X-ray Spectroscopy (EDS) on FIB Lyra 3 Tescan microscope.

### 3.3. Strains and Growth Condition

In the present study, we used reference strains of *Enterococcus faecalis* ATCC 29212 and *Staphylococcus aureus* ATCC BAA 1556 (USA300), and clinical isolates of *Enterococcus faecalis* 37VRE and *Staphylococcus epidermidis* P36. The bacterial strains were cultured in Mueller Hinton II Broth BD (MHB) (Oxoid, Nepean, Canada). The strains were incubated aerobically for 24 h at 37 °C. Overnight microorganism cultures were centrifuged, washed with PBS (pH 7.4), and suspended in fresh MHB to obtain suitable optical density [44].

### 3.4. Photocatalytic Inactivation Experiment

The overnight cultures of strains were centrifuged, collected, and diluted in a sterile PBS to a concentration of 1–3 × 10^5^ CFU (colony forming unit)/mL as determined from optical density measurements (OD600). One milliliter of bacterial suspension was then added to 1 mL of the nanopowder suspension placed in a glass dish, to give microcomposites a final concentration of 750 μg/mL and 1500 μg/mL. The dishes were placed on a stir plate with continuous stirring (to ensure maximal mixing and prevent settling of the nanoparticles) and illuminated using the xenon light above (the distance of 10 cm between the lamp and the dish; 50 mW·cm^−2^), (OPTEL lamp, Opole, Poland), for 60 min. The same set of samples was prepared for a parallel experiment conducted in total darkness conditions. At each time point (0 min, 30 min, and 60 min), 100 µL of suspension was removed from the dish (both from irradiated and not irradiated systems) and diluted from 10^−1^ to 10^−6^ CFU/mL. Quantification of the viable cells was performed by the standard plating method. To obtain between 30 CFU and 300 CFUs per plate, 50 μL of each dilution was inoculated onto Trypticase Soy Agar (TSA) (Oxoid, Canada) plates, in triplicate. Then, the plates were incubated for 24–48 h at 37 °C. Following incubation, colonies were counted and multiplied by the appropriate dilution factor to determine the original colony count. Results were reported in CFU/mL.

### 3.5. Leakage of Cellular Metabolites

A modified method by Paluch, et al., 2018 was used to determine DNA leakage [45], and Lin et al., 2000 for protein measurement [45]. The *E. faecalis* 37VRE culture was incubated in Lysogeny Broth (LB) (Oxoid, Canada) for 18 h at 37 °C, respectively, with shaking (140 rpm). The bacteria were then purified, rinsed with sterile water, and suspended in Milli-Q water until reaching 0.5 McF. The research samples prepared in this way were subjected to the photocatalysis procedure described earlier (XE series). Tested materials were used at a final concentration of 750 μg/mL. As a control for photocatalysis, samples that were not irradiated (series D) but treated with test compounds were used. The negative control (C−) was bacterial cells that had not been treated with the compounds (initially, 2 irradiated and non-irradiated controls were used, but due to the lack of significant differences, only the irradiation was shown. Positive control (C+) were cells treated with mureinase (30 min, 37 °C, 400 RPM) and 1% SDS. After the incubation time, all samples were centrifuged (2 min, 4 °C, 12,000 RPM) and the resulting supernatant was transferred to new Eppendorf tubes. The protein content of the supernatant was read under UV light λ = 280 nm using the Nanodrop 2000c spectrophotometer (Thermo Scientific, Waltham, MA, USA). Then, 400 µL of 5M potassium acetate was added to the samples to determine the amount of DNA and incubated for one hour on ice. The samples were centrifuged (15 min, 14,000 RPM, 4 °C) and the supernatant was transferred to new tubes. Then, 1 volume of ice-cold 100% 2-propanol was added and incubated for 12 h at −20 °C. The samples were centrifuged (15 min, 12,000 RPM, 4 °C) and the resulting DNA pellet was dried for 2 h at 24 °C in a vacuum dryer (Haraeus Megafuge) and suspended in 30 μL of Milli-Q water. The final concentration of DNA was measured in the light UV λ = 260 with the Nanodrop 2000c spectrophotometer (Thermo Scientific, USA). The differences in the absorption values at 260 nm and 280 nm between controls and test groups were used to estimate cellular macronutrient (protein) leakage. The experiment was repeated three times.

### 3.6. Leakage of Potassium Ions

The modified Duan, et al., 2017 method was used to determine the leakage of potassium ions from *E. faecalis* 37VRE cells [46]. From the previously prepared culture, a bacterial suspension with a density of 0.5 McF was prepared in sterile Milli-Q water. Then, the photocatalysis procedure was performed analogously. Tested materials were used at a final concentration of 750 μg/mL The negative control was cells suspended in 2 mL of sterile Milli-Q water, and the positive control was autoclaved cells (30 min, 1 atm, 121 °C). After this time, all samples were centrifuged (2 min, 14,000 RPM), and the supernatant was analyzed by ICP-OES analysis for potassium ion content according to the procedure described below in section [47].

### 3.7. Penetration of Silver and Titanium Ions

A suspension of 0.5 McF in sterile Milli-Q water was prepared from the *E. faecalis* 37VRE culture. Then, photocatalysis was performed according to the above-described procedure. Tested materials were used at a final concentration of 750 μg/mL. After photocatalysis, cells were purified of possible residues of test compounds by centrifugation repeated 3 times (30 s, 5000 RPM), followed by suspending the cells in 2 mL of sterile Milli-Q water. Purified cells were then re-centrifuged (10 min, 5000 RPM, 4° C), suspended in 2 mL of sterile Milli-Q water and autoclaved (30 min, 1 atm, 121 °C), cell debris was centrifuged (15 min, 12,000 RPM, 4 °C), and the supernatant was transferred to new tubes and subjected to ICP-OES analysis for silver and titanium ions, according to the parental procedure described below [48]. 

### 3.8. Ions Content Analysis ICP-OES

Ions content analysis was performed according to the procedure of Szymczycha-Madeja et al., 2019 [48]. A bench-top optical emission spectrometer, model 720 (Agilent, CA, USA), with an axially viewed Ar-ICP and a 5-channel peristaltic pump, was used to measure the concentrations of trace elements. The instrument was equipped with a high-resolution echelle-type polychromator and a Vista Chip II CCD detector (Agilent) cooled down to −35 °C on a triple-stage Peltier device. The plasma was sustained in a standard 1-piece, low-flow, extended quartz torch with a 2.4 mm inside diameter injector tube. A single-pass glass cyclonic spray chamber and a OneNeb pneumatic concentric nebulizer made of high-tech PFA and PEEK polymers were used to introduce the sample solutions by pneumatic nebulization. Operating conditions recommended by the manufacturer for solutions containing high levels of dissolved solids were applied: an RF power of 1200 W, a plasma gas flow rate of 15.0 L/min, an auxiliary gas flow rate of 1.5 L/min, a nebulizer gas flow rate of 0.75 L/min, a sample flow rate of 0.75 mL/min, a stabilization delay of 15 s, a sample uptake delay of 30 s, a rinse time of 10 s, a replicate read time of 1 s, and 3 replicates. A fitted background mode with 7 points per line profile was applied for the background correction. Background-corrected intensities of analytical lines were used for calibration graphs.

### 3.9. Oxidative Stress and ROS Production

Two complementary methods were used for the analysis of oxidative stress–quantitative analysis (spectrofluorimetric measurement of fluorescence units on 96-well titration plates) and observation of cells by fluorescence microscopy.

ROS production was examined according to Kim et al., 2011 [49]. *E. faecalis* 37VRE was incubated in the LB medium for 20 h at 37 °C with shaking at 140 RPM, and next the culture was diluted to a density of 0.5 McF. Then, the photocatalysis procedure described earlier was performed analogously. Tested materials were used at a final concentration of 750 μg/mL The cells were then purified by 3× centrifugation (30 s, 5000 RPM) by suspending the cells in 2 mL of sterile milli-Q water. The negative control consisted of cells suspended in 2 mL of sterile Milli-Q water, and the positive control was cells suspended to a density of 0.5 McF in 1.5% perhydrol (H_2_O_2_). Cells were centrifuged (4000 RPM, 5 min), suspended in 2 mL lysis buffer (50 mM HEPES, 2 mM PMSF, protease inhibitor cocktail; pH 7.5), then cells were digested with mureinase (30 min, 37 °C, 400 RPM) and lysed using a sonication (Bandelin SonoPuls HD 2070), (power 30%, 3 cycles for 1 min). Cell debris was centrifuged (10 min, 5000 RPM, 4 °C) and the protein amount was measured λ = 280 nm using a UV–VIS Nanodrop 2000c (Thermo Scientific, USA). The fluorescence of the supernatant was measured using a spectrofluorimeter multimode microplate reader Varioskan LUX (Thermo Fisher Scientific, USA) utilizing a microplate 96-well PS F-bottom, black (CHIMNEY WELL). Results were normalized to the protein levels. Cells were stained with fluorescent markers at a final concentration of 5 µM for 30 min in the dark. The following parameters were used for fluorescence measurements: 2ʹ,7ʹ-dichlorodihydrofluorescein diacetate (H_2_DCFDA) (λex = 495 nm, λem = 517 nm), dihydroethidium (DHE) (λex = 460 nm, λem = 640 nm) and hydroxyphenyl fluorescein solution (HFS) (λex = 495 nm, λem = 517 nm). The fluorescence of cells was analyzed by fluorescence microscopy compared to a bright field (100×) using an Olympus BX51 with DP25 camera (Olympus, Tokyo, Japan). Scale bar = 10 μm.

### 3.10. Bacterial Viability (CLSM)

The experiment used a modified procedure of Manteca et al., 2005 [50]. After the photocatalysis procedure described previously, cells were then stained for 30 min in the dark with the LIVE/DEAD ™ BacLight ™ Bacterial Viability Kit to determine the viability of bacterial cells treated with test compounds. Tested materials were used at a final concentration of 750 μg/mL. The negative control (C−) was cells suspended in 2 mL of sterile Milli-Q water, and the positive control (C+) was cells suspended to a density of 0.5 McFarland standard in 1.5% perhydrol (H_2_O_2_). Cells were imaged with an Olympus IX83 Fluoview FV 1200 confocal microscope with a Hamatsu C13440 CCD camera at 40× magnification and the following laser lines: propidium iodide (Ex λ = 543 nm), Syto 9 (Ex λ = 488 nm). Scale bar = 100 μm. Acquired images were processed and analyzed in the software Fiji/ImageJ software ver. 1.53c (NIH). First, maximum intensity projections (MIP) were obtained from stacks of images. The areas from binarized images were then transferred onto original live and dead channel MIP images and mean fluorescence intensities of all detected objects per field of view were calculated using the ImageJ’s Analyze Particles function.

### 3.11. TEM and SEM EDS Analysis

*E. faecalis* 37VRE cells were suspended in sterile water mili-Q to obtain 0.5 McFarland and centrifuged (5 min, 12,000 RPM). The resulting material was incubated for 8 h in 4% glutaric aldehyde (pH 7.4), then washed for 24 h in 0.2 M PBS (pH 7.4), and preserved for 2 h in 2% osmium oxide (VIII). Cells were then centrifuged (5 min, 3000 RPM) and washed for 30 min in redistilled water. Clean samples were dried in 50%, 70%, 80%, 90%, 96%, and 100% alcohol–acetone series (three times) with 15 min incubation for each of the solutions. Dehydrated cells were suspended in acetone:Epon 812 mixture (1:1) and incubated for 16 h. Saturated cells were then immersed in Epon 812 epoxy and the polymerization was carried out for 24 h at 45 °C and 60 °C. Epoxy semi-droplets were cut into semi-thin sections and ultrathin sections with a diamond knife (Reichert-Jung, Depew, NY, USA). Ultrathin sections were treated for 15 min with 2% solutions of uranyl acetate and lead citrate for contrast and observed under the transmission electron microscope (TEM) Tesla BS-540 [51]. Cells untreated with tested compounds were used as a control. In parallel, selected research samples (AGP, AGP/P25, HA/AGP XE series–VIS light exposed) were prepared for SEM EDS analysis, and ultra-thin sections were placed on foil carbon grids and analyzed according to the procedure of Zielonka et al., 2018 [52]. SEM images and chemical composition of the samples were checked with a FE-SEM microscope (FEI Nova NanoSEM 230, Oregon, USA) equipped with an EDS analyzer (EDAX Genesis XM4). SEM and EDS measurements were carried out using an accelerating voltage of 5.0 and 10.0 kV, respectively.

### 3.12. Statistical Analysis

In this work variance analysis was performed using the software Statistica (ANOVA analysis). Results for which *p* < 0.05 were treated as significant (indicating that the results are not identical at a 95% confidence level). All experiments were performed three times and at least in triplicate. Data are expressed as means ± standard deviation.

## 4. Conclusions

In this work, we demonstrated the antibacterial potential of Ag_3_PO_4_ photocatalyst and its composites with titanium dioxide P25 and hydroxyapatite. The photocatalytic inactivation process brings new disinfectant opportunities in public environments and hospital settings, which are ideal places for the transmission of a wide range of harmful microorganisms relevant to hygiene, such as *Enterococci*, *Staphylococci*, and other bacteria. Application of this technology for the disinfection of contaminated surfaces and inanimate objects can help to reduce the risk of spread of the pathogens. Photocatalytic materials based on silver phosphate have promising potential for use in the production of, e.g., medical implants, surgical tools or self-disinfecting equipment in the hospital and laboratory settings. The use of innovative microcomposites brings the possibility of using them in biofilm prevention. Generally, our studies have shown that Ag_3_PO_4_-based composites are highly effective agents that can eradicate 100% of bacterial populations during the 60 min photocatalytic inactivation. At lower doses, the activity of the tested compounds depends on the type of pathogen, the duration of exposure to VIS light, and many other factors. However, despite these differences, the tested composites should be considered an effective antimicrobial agent that requires further research, but has great potential. 

All these facts make this work deserving of attention:It shows a detailed antibacterial mechanism of action of the investigated microcomposites;Studies on the antimicrobial activity of the tested compounds involved resistant strains of Gram-positive bacteria, both reference and clinical with a great impact on nosocomial infections;Compared to other studies (generally focused on one compound), these presented in the following article include the antimicrobial activity of several different silver phosphate composites exposed to VIS light or without exposure.

## Figures and Tables

**Figure 1 ijms-24-13553-f001:**
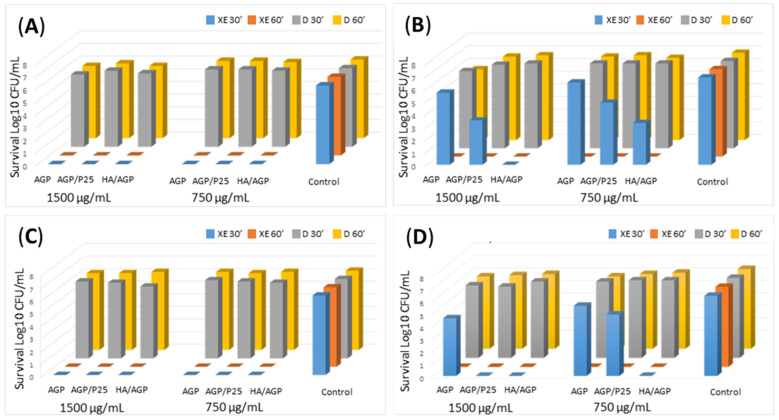
Inactivation of Gram-positive bacteria on Ag_3_PO_4_-based materials: (**A**) *E. faecalis* 37VRE; (**B**) *E. faecalis* ATTC 29212; (**C**) *S. aureus* ATCC 1556 (USA300); (**D**) *S. epidermidis P36*. Xe series—30′, 60′—exposure to VIS light for 30 min and 60 min; D series—non-exposure to VIS light; 750 mg/mL, 1500 mg/mL—final concentration of tested materials (AGP–Ag_3_PO_4_, AGP/P25, HA/AGP).

**Figure 2 ijms-24-13553-f002:**
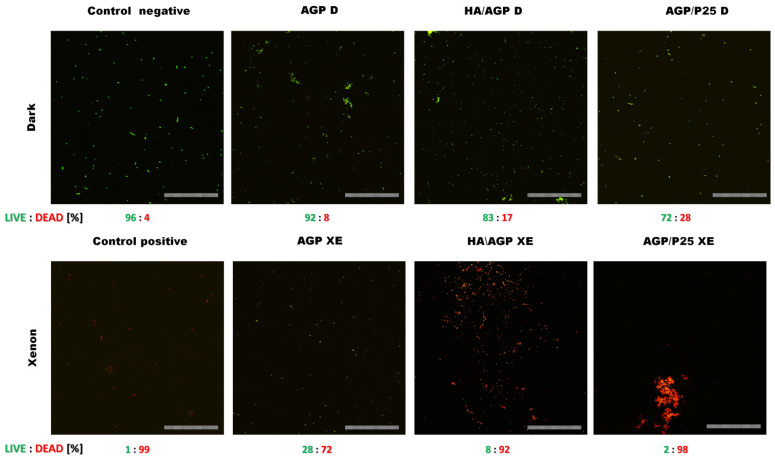
*E. faecalis* cells exposed to tested compounds (top in dark (D)) and (bottom after photocatalysis (XE)), stained with propidium iodide (PI) (red fluorescence), and Syto 9 (green fluorescence). Cells not exposed to the activity of the test compounds were a negative control. The positive control was bacterial cells exposed to 1.5% hydrogen peroxide (H_2_O_2_). Scale bar = 100 μm.

**Figure 3 ijms-24-13553-f003:**
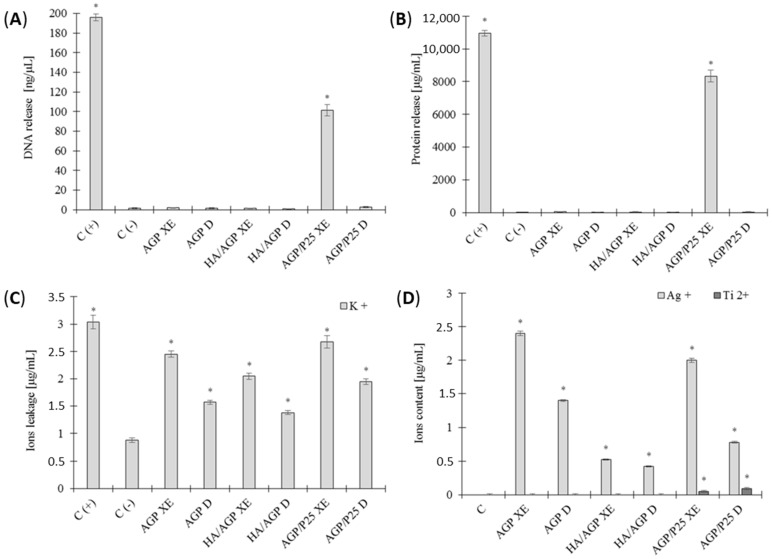
Leakage of intracellular metabolites from *E. faecalis* cells: (**A**) DNA, (**B**) proteins, (**C**) potassium ions (K^+^), and (**D**) the content of metal ions (Ag^+^) and (Ti^4+^) in cells after photocatalysis. The positive control (C+) for the leakage of DNA and protein (**A**,**B**) were protoplasts exposed to 1% SDS, and for the leakage of potassium ions autoclaved cells. Negative control (C−) were cells not exposed to compounds; ±SD, n = 3; * statistically significant (*p* < 0.05). ”XE” series—exposed to light VIS, “D” series—non-exposed.

**Figure 4 ijms-24-13553-f004:**
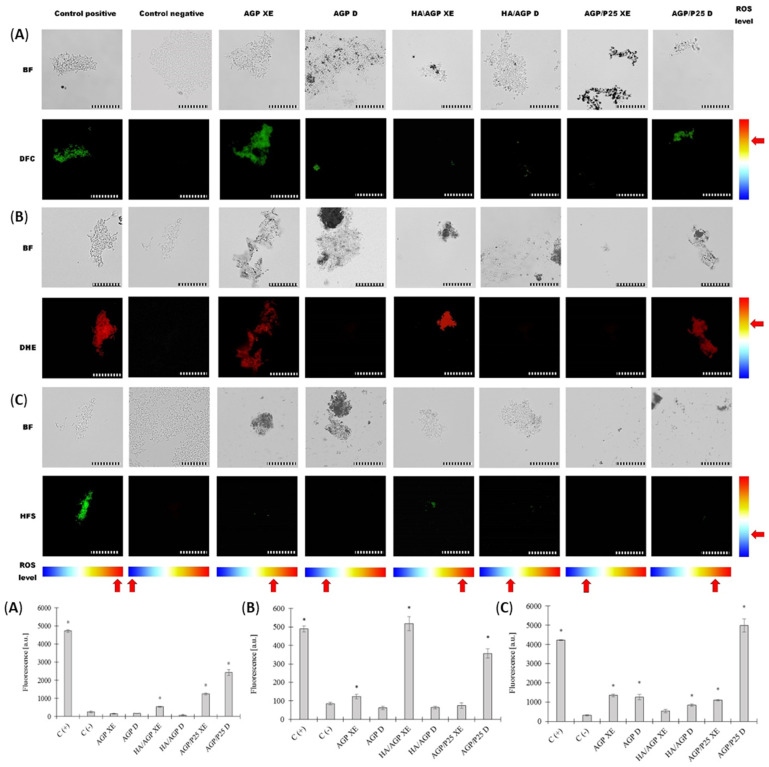
Microscopy: bright field (BF) and (DFC, DHE, HFS) fluorescence microscopy (**top**). Photocatalyzed E. faecalis cells showing: (**A**) general oxidative stress (DCF); (**B**) increased production of superoxide radical (O_2_^•−^) (DHE); and (**C**) production of highly reactive ROS, including hydroxyl radical anion (HO^•^); measuring bar = 10 μm. Spectrofluorometric measurement of the units of fluorescence intensity of oxidative stress, respectively: (**A**) DCF, (**B**) DHE, and (**C**) HAF (**bottom**). Negative control (C−) bacterial cells untreated with compounds. Positive control (C+) bacterial cells exposed to 1.5% hydrogen peroxide (H_2_O_2_); mean ± SD, n = 30; * statistically significant *p* < 0.005.

**Figure 5 ijms-24-13553-f005:**
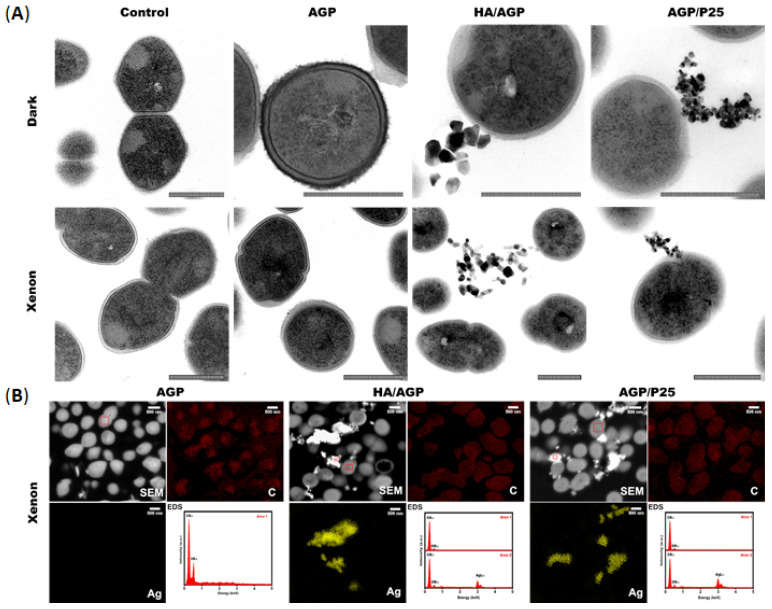
*E. faecalis* 37VRE cells were imaged using: (**A**) transmission electron microscopy (TEM) (on the top non-exposed “dark” and at the bottom photocatalyzed–Xenon) with visible nanoparticles in the case of HA/AGP and AGP/P25 compounds and (**B**) selected samples subjected to scanning electron microscopy with energy-dispersive X-ray spectroscopy (SEM EDS) analysis with detection (spectra) and visualization of carbon and silver atoms in the tested material. The control was cells not exposed to compounds, bar = 500 nm (TEM).

**Figure 6 ijms-24-13553-f006:**
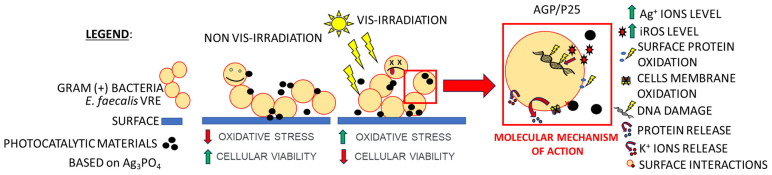
The molecular mechanism of action of the tested photocatalytic materials based on Ag_3_PO_4_ (especially AGP/P25) against Gram-positive bacteria in particular *E. faecalis*.

## Data Availability

Not applicable.

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
