# Peer review of "Mechanism of Action and Efficiency of Ag3PO4-Based Photocatalysts for the Control of Hazardous Gram-Positive Pathogens"

_ijms, 2023, doi:10.3390/ijms241713553_

Round 1
Reviewer 1 Report

Moderate editing of English language is required
Author Response
Dear Reviewer,
we send response in attached.

Reviewer 2 Report
This manuscript, submitted for consideration for publication in IJMS, by Paluch et al. presents an investigation on the antimicrobial properties of silver phosphate (Ag3PO4 - AGP) composites with titania and hydroxyapatite (HA) on three strains of gram-positive bacteria.
A detailed characterization is introduced, where the antimicrobial activity is discussed from the angles of: (1) general bacterial inactivation in darkness, and under visible light illumination; (2) viability of the bacteria via confocal laser scanning microscopy; (3) leakage of intracellular compounds - proteins, DNA, K+ and catalyst-associated ions; (4) oxidative stress factors, associated with reactive species; and (5) general morphology of the cells via microscopic techniques; The authors conclude that the combined photocatalytic action of AGP/TiO2 and, according to data AGP/HA, composites has an enhancement effect on the inactivation of the strains studied.
Generally, the paper is of good quality, has a solid scientific basis, based on the combination of multiple techniques. The English quality is satisfactory and, as the background of the main author is in microbiology, it has a strong experimental planning with the introduction of multiple control experiments, which is pleasant (except for excluding controls with TiO2 or HA on their own), when compared to other papers dealing with photocatalytic research.
I minor deficit, that I sense, however, is that due to the above mentioned factor, there is a bit of lack of understanding on the photocatalytic mechanisms, which leads to some minor misinterpretations when it comes to photocatalysis - as the text mentions the formation of heterojunctions leading to enhanced activity in some places, usually related to TiO2, and I was a bit curious if some of the discussion can be expanded on topics as whether this mechanism will be relevant, given that the highest deactivation in the XE 30’ case is observed for the AGP/HA composite in all cases, and HA itself has been modestly explored as a photocatalyst, compared to TiO2, but there seems to be evidence that it has such properties.
Apart from this smaller issues I have, I would like to strongly recommend this paper for publication in IJMS, as it provides an interesting angle on photocatalytic inactivation of bacteria, with a strong microbiological background and would be helpful to other researchers working in photocatalysis. The only reason I recommend Minor revisions, is due to the lack of focus in some sections, and some lacking in the quality of the presentation. Here is list of a few points, expressed in detail about some changes and recommendations to the authors:
1) Please introduce hydroxyapatite (HA) abbreviation in the introduction. Also in the text. During my first read it was struggling to find what the abbreviation HA implies, and one could imagine that this would be the experience of any reader (in fact, the word “hydroxyapatite” is used explicitly, w/o being abbreviated, only on four occasions in the entire text, and HA is mentioned once as “HAP”.
In general, the paper has an issue with abbreviations that are not introduced. The abstract also has - ROS (reactive oxygen species); TEM, STEM EDS, P25 which is known by people in photocatalysis, but should also introduced that it stands for the commercial P25 anatase/rutile photocatalyst, etc. I strongly suggest the authors go carefully through the text and make sure to clarify any abbreviated, or notational term, is properly introduced to the reader for clarity. MDPI has a clear list of abbreviations that can be used without introduction, which is available in the Authors’ guidelines.
2) I actually realised that the authors use the term “P25” interchangeably with TiO2. Yes, P25 is TiO2, but it is the commercial form offered by company Degussa (which is nowadays called Evonik). Even checking an arbitrary reference, such as “Antimicrobial activity of P25 material was also demonstrated on orthopedic implant surfaces where they limited the growth of E. hirae [21].”, in Ref. 21, which is a work by Tsuang et al., they actually do not use Degussa P25, but another commercial TiO2 powder from a Japanese company (ST-01). While P25 is a benchmark material almost colloquial with photocatalytic TiO2 in literature, the term P25 explicitly means that it is the 75 wt.% anatase:25 wt.% rutile powder produced and offered by Degussa/Evonik.
I know that this remark is bordering with being “petty”, but to avoid confusion I strongly suggest that “TiO2” is used instead of “P25”, unless when referring to cases where the P25 catalysts is really used.
3) It would be helpful if the hole transfer mechanism between AGP and TiO2, discussed in the Introduction (the paragraph “In turn, valence band level of silver orthophosphate is lower …”) is presented schematically, along the corresponding energy diagrams.
Additionally, one of the following paragraphs (“In photocatalysis materials where electrons from the conduction band recombine with holes from the valence …”) presents an imprecise impression of the photocatalytic mechanism, describing something more akin to the recombination process. The ROS generation would imply transfer of electrons from/to nearby species - i.e. generating superoxide radicals by photoreduction or hydroxide radicals by photooxidation. These processes can simply be illustrated in an energetic scheme with many examples in the literature to help readers unfamiliar with photocatalysis.
4) I also have a very minor point regarding the mention of the cytotoxic effect of the nanoparticles itself. Indeed, it is a valid point, however, do the authors consider its practicality in applications ? There is actually a an EU regulation (Commission Regulation (EU) 2022/63) banning the use of TiO2 in consumer products only due to the possibility of nanoparticle cytotoxicity (it was widely reported in news, albeit, wrongly).
5) There are some typos in the text, such as “photocatalites" (photocatalysts); Additionally - STEM implies “Scanning Transmission Electron Microscope”, which is a mode of operation offered in some TEM microscopes, while according to the Methods section Scanning Electron Microscope is used. So it is incorrect to state STEM EDS, when it is SEM-EDS. Unless a transmission accessory is used in the SEM-EDS setup.
6) Page 3 is a full page with discussion on the results shown in Figure 1. However, it also contains a lot of references to literature studies, which are effectively “diluting” the interpretation offered on the experimental results. I commend the authors for the in-depth literature search to provide connections with other researchers’ work, however, I would recommend that the text is re-arranged there to accentuate on their findings and clearly show where the comparison with other studies begin. Otherwise it makes the text difficult to follow (and besides, a lot of the examples in Page 3 would have been better fit in the Introduction)
7) The type and power of the Xenon lamp is not defined in the text.
8) Figure 2 caption states, that the positive control implies the addition of 1.5% hydrogen peroxide, but the reasoning of this positive control is not outlined in the text before this figure.
9) On multiple occasions the text discusses “Ti2+” ions, which is a bit unnerving, given that Ti4+ is the most stable oxidation state, and the one in TiO2. Of course, reduced titania forms exist, and are not uncommon even for “black titania”, but it is an unfeasible expectation that Ti2+ would be released, even if there is photo dissolution of the material. The two studies mentioned [30, 31] have nothing to do with Cu2+, Zn2+, nor Ti2+ and only deal with noble metal ions. Finally, the detection of Ti by ICP-OES does not yield any information about presence of Ti2+ ions inside the cell. Previous studies have shown that there is uptake of the particles themselves, and if Ti is detected by the ICP-OES analysis, it’s probably due to particles and not ions inside the cell (P25 has a size of 20 nm). Of course, I must state, that this criticism is not directed at the validity of the results, and only on small part of their interpretation.
There are just some minor typos, and punctuation mistakes, that will be surely spotted upon a careful read-through of the final version. Also, pay attention to the reference brackets, as some of them include extra commas, as if the next reference number is missing.
Author Response
Dear Reviewer,
we send response in attached.
Special thank you for your work and suggestions.
Regards

Reviewer 3 Report
In the present paper, the Authors described the mechanism of action and efficiency of Ag3PO4-based photocatalysts for the control of hazardous Gram-positive pathogen. The paper is probably publishable, but needs a revision before any further consideration for publication. The main concerns include:
1. The elements of novelty should be more emphasized in the introduction.
2. The information about the lamp used in the experiments should be more detailed. What is the range of irradiation emitted by this light source? The intensity of light should be also given.
3. Conclusions part should be more informative. This part is too general. There is also no information about which material is the most promising.
The quality of English language is satisfactory. Only minor editing of English language is required.
Author Response

(The authors gave the same response as above.)
